

# Three-year monitoring of stable isotopes of precipitation at Concordia Station, East Antarctica

Barbara Stenni[1], Claudio Scarchilli[2], Valerie Masson-Delmotte[3], Elisabeth Schlosser[4,5], Virginia
Ciardini[2], Giuliano Dreossi[1], Paolo Grigioni[2], Mattia Bonazza[6], Anselmo Cagnati[7], Daniele Karlicek[8],
Camille Risi[9], Roberto Udisti[10], Mauro Valt[7]

[1]Department of Environmental Sciences, Informatics and Statistics, Ca' Foscari University of Venice, Italy

[2] Laboratory for Earth Observations and Analyses, ENEA, Rome, Italy

[3]LSCE (UMR 8212 CEA-CNRS-UVSQ/IPSL), Université Paris Saclay, Gif-sur-Yvette, France

[4] Institute of Atmospheric and Cryospheric Sciences, University of Innsbruck, Innsbruck, Austria

[5]Austrian Polar Research Institute, Vienna, Austria

[6]Faculty of Forest Sciences and Forest Ecology, Bioclimatology Department, Georg-August-Universität Göttingen, Germany

[7] ARPA Center of Avalanches, Arabba, Italy

[8] Department of Mathematics and Geosciences, University of Trieste, Trieste, Italy

[9] Laboratoire de Météorologie Dynamique, Paris, France

[10] Department of Chemistry "Ugo Schiff", University of Florence, Florence, Italy

*Correspondence to: B. Stenni (barbara.stenni@unive.it)*



**Abstract.** Past temperature reconstructions from Antarctic ice cores require a good quantification and understanding of the relationship between snow isotopic composition and 2 m air or inversion (condensation) temperature. Here, we focus on the French-Italian Concordia Station, central East Antarctic plateau, where the European Project for Ice Coring in Antarctica (EPICA) Dome C ice cores were drilled. We provide a multi-year record of daily precipitation types identified from crystal

morphologies, daily precipitation amounts, and isotopic composition. Our sampling period (2008-2010) encompasses a warmer year (2009, +1.6°C with respect to 2 m air temperature period average), with larger total precipitation and snowfall amounts (14%, 76% above average, respectively), and a colder and drier year (2010, -1.4°C, 4% below average, respectively) with larger diamond dust amounts (49% above average). Relationships between local meteorological data and precipitation isotopic composition are investigated at daily, monthly and inter-annual scale, and for the different types of precipitation.

Water stable isotopes are more closely related to 2 m air temperature than to inversion temperature at all time scales (e.g. $R^2$=0.63 and 0.44, respectively for daily values). The slope of the temporal relationship between daily $\delta^{18}O$ and 2 m air temperature is approximately two times smaller (0.49‰/°C) than the average Antarctic spatial (0.8‰/°C) relationship initially used for the interpretation of EPICA Dome C records. In accordance to results from precipitation monitoring at Vostok and Dome F, deuterium excess is anti-correlated with $\delta^{18}O$ at daily and monthly scales, reaching maximum values in winter. Hoar

frost precipitation samples have a specific fingerprint with more depleted $\delta^{18}O$ (about 5‰ below average) and higher deuterium excess (about 8‰ above average) values than other precipitation types. These datasets provide a basis for comparison with shallow ice core records, to investigate post-deposition effects. A preliminary comparison between observations and precipitation from the European Centre for Medium-Range Weather Forecast (ECMWF) re-analysis and the simulated water stable isotopes from the Laboratoire de Météorologie Dynamique Zoom atmospheric general circulation model (LMDZiso),

shows that models do correctly capture the amount of precipitation as well as more than 50% of the variance of the observed $\delta^{18}O$, driven by large scale weather patterns. Despite a warm bias and an underestimation of the variance in water stable isotopes, LMDZiso correctly captures these relationships between $\delta^{18}O$, 2 m air temperature and deuterium excess. Our dataset is therefore available for further in depth model evaluation at the synoptic scale.

## 1 Introduction

Antarctic ice cores provide exceptional past climate records, thanks to the wealth of climatic and environmental information archived in the water and air of deep ice cores (e.g. Jouzel and Masson-Delmotte, 2010; WAIS Divide Project Members, 2015). Amongst these various proxies, water stable isotopes are integrated tracers of the atmospheric water cycle and local climate (Masson-Delmotte et al., 2008, 2011). Water stable isotope records from central Antarctica are pivotal for reconstructions of past temperature and mass balance (Parrenin et al., submitted). Low accumulation sites in the central Antarctic plateau provide

the longest continuous ice core records retrieved so far, documenting the last eight climatic cycles (Jouzel et al., 2007), with a potential to expand these records beyond one million years (Fischer et al., 2013). This motivates an improved knowledge of relationships between water stable isotopes and climate.



Since the 1950s, observations (Dansgaard, 1964), theoretical distillation models (Jouzel and Merlivat, 1984) and atmospheric general circulation models (Jouzel, 2014) have evidenced a close relationship between the isotopic composition of polar precipitation and condensation temperature. While this has formed the basis for past temperature reconstructions from deep Antarctic ice cores, key sources of uncertainties have been identified in the climatic interpretation of water stable isotope records.

Uncertainties arise from the empirical relationship between the isotopic composition of precipitation and temperature. This relationship is affected by atmospheric processes associated with fractionation processes and mixing occurring from evaporation to deposition, as well as through the intermittency of precipitation events, which provide an irregular sampling of specific weather events. In this perspective, it is expected that condensation temperature should be the key driver of distillation, and most closely related to the precipitation isotopic composition. Another source of uncertainty arises from post-depositional processes, as recent monitoring studies performed above the Greenland ice sheet have suggested that the isotopic composition of surface snow can change in-between snowfall events due to snow-atmospheric exchanges in relationship with surface snow metamorphism (Steen-Larsen et al., 2014). While there is recent evidence for water vapour exchange between surface snow and Antarctic air (Ritter et al., 2016; Casado et al., 2016, Touzeau et al., 2016) the importance of this process for ice core records has not yet been quantified.

Empirical relationships between surface snow isotopic composition and temperature were established based on the geographical (spatial) relationships between surface snow samples and the temperature obtained from weather stations or measured at 10 m depth in the firn, with sometimes additional assumptions on relationships between 2 m and condensation air temperature. In a compilation of available Antarctic data (Masson-Delmotte et al., 2008), spatial isotope-temperature slopes of 0.80 and 6.34‰/°C were calculated for $\delta^{18}O$ and $\delta D$, respectively, with an uncertainty of ±20% associated with the spatial variability of this relationship.

Alternatively, atmospheric general circulation models equipped with water stable isotopes have been used to investigate the relationships between the simulated precipitation isotopic composition and temperature for cold climate conditions, such as the Last Glacial Maximum, or extreme warm climate conditions, such as for increased $CO_2$ projections. While all existing simulations have suggested that the modern spatial relationship for central Antarctica is a good approximation for glacial to present day changes (Jouzel et al., 2003), it may not apply for warm climatic conditions. Schmidt et al. (2007), using a coupled ocean-atmosphere model, reported a variability in the temporal slope for $\delta^{18}O$ in East Antarctica of 0.2-0.5‰/°C for simulations performed at the inter-annual scale and during the mid-Holocene. Moreover, Sime et al. (2008) simulated a much weaker sensitivity of $\delta^{18}O$ to temperature (0.34‰/°C) together with significant regional differences in a simulated climate driven by increased atmospheric $CO_2$ concentration. This was attributed to changes in the covariance between simulated temperature and precipitation.

Finally, attempts have been made to account for changes in moisture sources, thanks to combined measurements of $\delta D$ and $\delta^{18}O$ allowing calculation of the second-order parameter deuterium excess (d= $\delta D$-8∗$\delta^{18}O$). The coefficient of 8 in this



definition accounts for the global mean ratio of equilibrium fractionation effects for oxygen and hydrogen isotopologues. In Antarctic precipitation, deuterium excess is partly affected by air mass distillation effects, due to the impact of condensation temperature on equilibrium fractionation (and deviation from this coefficient of 8), but also by changes in evaporation conditions, themselves driven by sea surface temperature, wind speed, and relative humidity at the sea surface (Uemura et al.,

2008) as well as kinetic effects during snow formation (Jouzel and Merlivat, 1984). Based on simulations performed with distillation models, it is possible to extract information on both condensation temperature and evaporation conditions from the combined measurements of water stable isotopes. This methodology has been applied to several deep Antarctic ice cores (e.g. Stenni et al., 2001; Vimeux et al. 2002; Uemura et al. 2012). This approach led to the conclusion that significant changes in evaporation conditions occurred at glacial-interglacial scales, but with secondary effects for the isotope-temperature

relationships.

This overview illustrates the incomplete understanding of the relationship between the isotopic composition of Antarctic snowfall and local temperature, which, so far, mostly relies on either mean spatial relationships, or on simulations performed with atmospheric general circulation models with known caveats for the Antarctic climate (Flato et al., 2013).

A better process-based understanding is expected from snowfall isotopic composition monitoring, from the scale of weather

systems to seasonal and inter-annual variations. Pioneering studies have been conducted in coastal Antarctica and on the central East Antarctic plateau, where deep ice cores have been retrieved (Fig. 1, Table 1).

For instance, Schlosser et al. (2004) used fresh snow samples collected after each snowfall event at Neumayer Station, situated in a coastal area. They investigated the relationships between snow $\delta^{18}O$ and temperature as a function of the synoptic situation and air mass trajectory classes, based on 5-day backtrajectories. The $\delta^{18}O$-temperature relationship was different for different

trajectory classes and strongly dependant on the presence of sea ice along air mass pathways.

Fujita and Abe (2006) used a 1-year dataset of daily precipitation measurements at Dome Fuji Station, on the East Antarctic Plateau to depict a strong linear relationship between stable isotope ratios and air temperature, coherent with the regional geographical relationship using 10 m-snow temperature. Ekaykin et al. (2004) reported a significant correlation between monthly values of $\delta D$ and air temperature for one-year snowfall sampling performed at Vostok. They obtained a temporal

slope significantly lower than the spatial one (Table 1).

Precipitation measurements are also important for evaluation of atmospheric circulation models in their ability to capture snowfall timing, amount and isotopic composition. So far, the ability to reproduce Antarctic isotopic signals is benchmarked mostly using spatial patterns (e.g. Werner et al, 2011; Risi et al., 2010). Obtaining high-resolution snowfall datasets is important to test if models correctly resolve weather-scale processes.

Here, we use the first and so-far only multi-year series of daily precipitation measurements and sampling from the French-Italian Concordia Station (Fig. 1), located at Dome C in East Antarctica (75°06'S 123°21'E; elevation: 3233 m a.s.l.; temperature at 10m: -54.5°C; snow accumulation rate: 25 kg m$^{-2}$yr$^{-1}$), where the longest continuous ice core records have been retrieved in the framework of the EPICA project (Jouzel et al, 2007). Snow particle observations provided an identification of precipitation types (diamond dust, hoar frost and snowfall), allowing us to explore potential impacts on their isotopic



composition. We also used our dataset and accumulation measurements to assess the capacity of one operational meteorological model to resolve accumulation, and one atmospheric general circulation model equipped with water stable isotopes to resolve isotope-temperature relationships at the daily to monthly scale. Section 2 describes the meteorological and precipitation isotopic composition data as well as the simulations explored in the model-data comparison. Section 3 reports

and discusses our results with a focus on the isotope-temperature relationship, and includes a model-data comparison. The implications of our results and outlooks are finally presented in Section 4.

## 2 Data and Methods

### 2.1 Temperature

Different automatic weather stations (AWS) have been operated around Concordia Station.  The longest time series stems from the US AWS, which was set up by the Antarctic Meteorological Research Centre of the University of Wisconsin-Madison in December 1995 at a distance of 1.5 km from Concordia (http://amrc.ssec.wisc.edu). The AWS provides 10 minute-averages of the respective meteorological variables, which were used to calculate the daily mean air temperature at 2-meter height ($T_{2m}$). Since 2005, vertical profiles of different parameters have been retrieved from radiosonde data, launched once per day by the

Institut Polaire Français Paul Emile Victor – IPEV / Italian Antarctic Meteo-climatological Observatory (http://www.climantartide.it).

A strong surface-based temperature inversion is occurring at Dome C which can reach a strength of up to 35°C in winter and is restricted to the lowest 100 m or less (Pietroni et al., 2014). As a first approximation, the condensation temperature (Jouzel and Merlivat, 1984, Masson-Delmotte et al., 2008) is assumed to be equal to the temperature at the upper limit of the inversion

layer. It is determined from daily radio sounding profiles as the bottom of the first layer where temperature decreases with altitude. Thin non-inversion layers are ignored if they are embedded within a deeper inversion layer (Kahl, 1990). We are aware that this approach is associated with large uncertainties, as small changes in height may lead to large temperature differences.

### 2.2 Precipitation data

Since the summer campaign of 2005-2006, precipitation samples have been manually collected year-round on a 80 x 120 cm wooden platform covered by a polystyrene/Teflon plate and standing 1 m above the snow surface. It is situated at a distance of about 800 m from Concordia Station. The layer of accumulated snow was collected at 1 am UTC (Coordinated Universal Time). This snow layer generally has a thickness varying from 0 to 10 mm. A few isolated cases of 30-50 mm thickness related

to blowing snow events were also observed. To prevent the snow from being blown off, the plate was shielded by a rail of 5 cm height. We cannot rule out impacts of winds on our sampling: snow may be blown onto (or off) the platform by the wind,





with or without precipitation. The sampling frequency has increased with time: since the end of 2007, sampling has been performed on a daily basis. We report here results for the period from 17 December 2007 to 31 December 2010, for a total of 1110 days and 607 precipitation samples.

When the amount of collected precipitation was too small to allow isotope analyses, the support plate was cleaned, no sample
was collected, and no precipitation amount was determined for that day. In all other cases, snow samples were immediately sealed into date-labelled plastic bags and preserved in a frozen state until their delivery at the Geochemistry Laboratory of the University of Trieste. Samples were then melted and transferred into appropriate HDPE (High-density polyethylene) vials and immediately stored into the freezers (≈-20°C) until their analysis.

Precipitation can consist of falling precipitation (snowfall and diamond dust) and deposited precipitation (hoar frost). Diamond
dust is sometimes called clear-sky precipitation, even though it can also be observed under a high cloud cover. It consists of very fine ice crystals that form due to radiative cooling of an almost saturated air mass. This can occur with or without the presence of condensation nuclei. The crystal type of diamond dust is a function of air temperature; fine needles are usually reported at temperatures below -40°C. Snowfall is observed when advection of moisture leads to cloudiness and "normal" precipitation, where characteristic snowflakes consist of an aggregate of ice crystals. Hoar frost forms when the surface is
cooled below the dew point of the overlying air, mostly due to radiative cooling. Various surfaces (e.g. ropes, flags, instruments, surface snow cover) then act as condensation surfaces.

Further information about the meteorological conditions was provided by the Concordia Base staff. The precipitation collected on our platform was examined daily using a magnifying lens and a high-resolution camera to determine these three broad types of precipitation (snowfall, diamond dust, and hoar frost) based on crystal morphology.

The surface snow accumulation has also been monitored at Dome C since 2005 as part of the GLACIOCLIM - Surface Mass Balance of Antarctica (SAMBA) observatory (http://www-lgge.obs.ujf-grenoble.fr/ServiceObs/index.htm). A 50-stake network was deployed near Concordia Station as a 1 km x 1 km cross and has been repeatedly surveyed for stake emerging length during each summer season. The network lies about 2 km upwind from the main station infrastructures. The uncertainty can be estimated through the variance of the 50 readings within a stakes network. It reaches about 10% of the annual mean
value. These snow height variations are converted into accumulation in water equivalent (w.e.) using a mean snow density value of 320 kg m$^{-3}$ following Genthon et al. (2015).

### 2.3 Stable isotope data

Stable isotope ratios of the precipitation samples were measured by isotope ratio mass spectrometric (IRMS) techniques
(Thermo-Fisher Delta Plus Advantage) using the well-established $CO_2$-$H_2$/water equilibration method. When the sample amount was less than 5 ml, the measurements were performed using a cavity ringdown spectroscope (CRDS) from PICARRO (model L1102-i), which only requires 0.3 ml amount of water. Measurements are reported against the Vienna-Standard Mean Ocean Water (V-SMOW) international standard. The IRMS provides an analytical precision of ±0.05‰ for $\delta^{18}$O and ±0.7‰





for δD, while the CRDS used here warrants a precision of ±0.1‰ for $\delta^{18}O$ and ±0.5‰ for δD, with a final precision on the calculated deuterium excess of ±0.8‰ and ±0.9‰, respectively.

## 2.4 Model data

The high-resolution precipitation measurements are used to test the ability of a global weather forecast model to produce realistic snowfall events driven by the synoptic scale weather. Simulated snowfall (SF) and evaporation (E) are obtained from the ERA-Interim Reanalysis, which is provided by the European Centre for Medium-Range Weather Forecast. Time series for Concordia Station are extrapolated from the 24-hour forecasted precipitation and evaporation fields, with a regular 0.75°x0.75° latitude-longitude grid, at the grid model point nearest to the Concordia site coordinates. Atmospheric models (both global and

mesoscale) systematically underestimate East Antarctic plateau precipitation, probably because clear-sky precipitation is not adequately parameterized (Bromwich et al., 2004; Van de Berg et al., 2006). However, Genthon et al. (2010) showed that part of these mismatches may, however, also arise from data inaccuracies. The ECMWF model does not account for blowing snow transport/sublimation. However, since the Concordia site is not influenced by strong winds this is expected to have minor importance here.

In order to test the suitability of the new Concordia dataset to evaluate isotopic-enhanced atmospheric general circulation models (GCM), we use here LMDZiso, the isotopic version (Risi et al., 2010) of the LMDZ4 atmospheric GCM (Hourdin et al., 2006), run at a resolution of 2.5°x3.75°, following the AMIP (Atmospheric Model Intercomparison Project) protocol. The simulation is nudged to the large-scale ECMWF operational analyses atmospheric circulation fields. The model has systematic caveats for the Antarctic climate, such as a warm and wet bias leading to a weaker $\delta^{18}O$ depletion than in the measurements

(Risi et al., 2013). We extracted the model daily outputs at the Concordia grid point.

## 3 Results

### 3.1 Observed and simulated variability of temperature and stable isotopes

In Figure 2, meteorological variables and stable isotope data of the fresh snow daily samples from Concordia Station are

displayed for the study period 2008-2010, together with their monthly means. The daily mean 2m-air temperature varies from about -80°C in winter to -25°C in summer (Fig 2a). Its annual course reflects the coreless winter typical for the interior of Antarctica (van Loon, 1967). During the polar night, with the lack of shortwave radiation, an equilibrium of downwelling and upwelling long-wave radiation is reached, leading to this "coreless winter". By contrast, a sharp summer temperature maximum follows the peak of insolation, at the end of December / early January.

The 2008-2010 sampling period encompasses remarkable inter-annual variations of atmospheric boundary layer (ABL) conditions at the Concordia site. In particular, the 2009 winter (June-July-August) was 3.6°C warmer than average (-62.2°C,




1996-2014 seasonal winter average), and a new record-high temperature was reached in July 2009. By contrast, the winter 2010 was 4.9°C colder than average and a new negative temperature record was observed in July 2010 (8°C below average). These contrasting conditions are related to an enhanced frequency of intrusions of warmer and moister air masses from lower latitudes in the 2009 winter (Genthon et al. 2013). While a strong zonal atmospheric flow was dominant in 2010, an enhanced

meridional flow prevailed in 2009, which increased the meridional transport of heat and moisture onto the East Antarctic plateau (Schlosser et al., 2016). This led to a number of precipitation/warming events at Concordia.

The ABL at Concordia is characterized by a persistent and generally strong inversion, which only disappears in summer in the early afternoon due to convective mixing (Genthon et al., 2010). During the study period, the inversion layer showed a large variability in both strength and vertical extension, with $T_{inv}$ reaching from -40°C in the middle of winter to -25°C in summer.

Temperature differences between 2 m and the top of the inversion layer, calculated from AWS and radiosonde data, respectively, can vary from 1-2°C in summer to 20-30°C in winter (Genthon et al 2013, Pietroni et al. 2014).

A linear regression of $T_{inv}$ as a function of $T_{2m}$ reveals a slope of 0.35 ($R^2$=0.67, n=1002) varying from 0.51 in summer (November - February, $R^2$=0.68, n=345) to 0.39 in winter (March - October, $R^2$=0.34, n=657), respectively. This is not in line with the simple empirical equation reported by Jouzel and Merlivat (1984) that yields a value of 0.67, neither with a 0.65 value

obtained from ERA-40 (1980–2002) relationship between annual surface temperature and weighted annual mean condensation temperature (Masson-Delmotte et al., 2008). However, these discrepancies may arise from the different methodologies to estimate $T_{inv}$ as well as the choice of different periods to calculate this relationship.

The $\delta^{18}O$ (Fig. 2a) and $\delta D$ (not shown) values range from -80.60 to -35.46‰, and from -584.7 to -284.0‰, respectively. The lowest $\delta^{18}O$ and $\delta D$ values found in daily snow samples over the whole period, both recorded on July 22[th], 2010, are close to

the isotopically lightest water ever collected on Earth (at Dome Fuji -81.9 and -595.5‰, respectively; Fujita and Abe, 2006). Both isotopes exhibit large seasonal amplitudes of the order of about 37‰ for $\delta^{18}O$ and 245‰ for $\delta D$ (average values from 2008, 2009 and 2010). The intra-seasonal variability of $\delta^{18}O$ appears more parallel to that of $T_{2m}$ in winter than in summer. At the inter-annual scale, the seasonal variations are larger in 2010 than in 2009.

The LMDZiso model has a warm bias at Concordia, with a mean annual simulated $T_{2m}$ value of -46.4°C, compared with our

observations of -51.15°C in 2008-2010 period. This warm bias is associated with too high $\delta^{18}O$ in all Antarctica (Risi et al., 2013). Despite a warm bias and too high $\delta^{18}O$ values, LMDZiso is able to capture some of the observed daily ($R^2$=0.50) to monthly ($R^2$=0.75) variability of $\delta^{18}O$ (Table 2), but underestimates the magnitude of $\delta^{18}O$ variability. LMDZiso is able to reproduce the observed daily ($R^2$=0.76) to monthly ($R^2$=0.90) variability of $T_{2m}$ (Table 2), but it is overestimating the magnitude of $T_{2m}$ variability. This is an encouraging result, given the low resolution of this atmospheric model, suggesting

that the nudging to winds from reanalyses drives a realistic synoptic variability.

The observed local meteoric water line is equal to $\delta D = 6.5\ \delta^{18}O - 68.8$ ($R^2$=0.98), considering all the precipitation values. The decrease of the $\delta D/\delta^{18}O$ slope from coastal to inland Antarctica is reflected on the deuterium excess calculation.



The deuterium excess daily observations (Fig. 2b) range from -32.19 to 60.11‰. Deuterium excess appears anti-correlated with $\delta^{18}O$ or $\delta D$. It is maximal in winter and minimal in summer (frequently reaching negative values), with an overall seasonal amplitude of about 70‰. The deuterium excess exhibits highest values of up to >60‰ in the cold winter of 2010, whereas in 2009 they stay mostly below 40‰. From daily data, deuterium excess is anti-correlated with $\delta^{18}O$ ($R^2$=0.70, n=499) with a

5 slope of -1.5‰/‰ (Table 2). This confirms the antiphase between $\delta^{18}O$ and deuterium excess also observed in precipitation samples from Vostok (Ekaykin et al., 2004) and Dome F (Fujita and Abe, 2006). Such anti-correlation is not observed in coastal Antarctica as at Neumayer Station in Dronning Maud Land and at Law Dome (Delmotte et al., 2000; Schlosser et al., 2008).

Despite the LMDZiso systematic bias and an underestimation of the variance of deuterium excess, the model is able to capture

this antiphase but with a weaker correlation coefficient ($R^2$=0.43), and a three times weaker slope than observed for both daily and monthly values (Table 2). This small deuterium excess - $\delta^{18}O$ slope is expected to be a direct consequence of the model warm bias, also evidenced in the model meteoric water line slope (7.5 ‰/‰).

## 3.2 Observed versus simulated variability of precipitation and accumulation

We now compare the cumulative accumulation from the stake array, the measured cumulative precipitation amount (Fig. 2c) with simulations of snowfall or accumulation (snowfall minus evaporation) from ECMWF and snowfall from LMDZ. Stake data are only available in the respective summer months.

The annual amounts of snow precipitation are 12.5, 15.7 and 13.3 mm water equivalent (w.e.) for 2008, 2009 and 2010, respectively. Taking into account the uncertainties associated with measurements of such extremely small amounts of

20 precipitation, the estimated precipitation amounts are coherent with the year-to-year variability of air temperature: larger amounts in winter 2009 (unusually warm and wet) and smaller amounts in 2010 (extremely cold and dry). From these two years, the relationship between precipitation amount and $T_{2m}$ is of 6.4% per °C, very close to climate model projection results (Frieler et al., 2015).

Our dataset also shows a much larger number of snowfall events (111 with respect to 31) in 2009 than in 2010 (Fig. 3). In

2008-2009, the precipitation amount has a minimum in late spring - early summer (NDJF) and a maximum in late autumn - winter (MJJA). However, the seasonality is different in 2010, where most of the precipitation occurred in late summer and early autumn (FMA) as diamond dust. A high inter-annual variability of both amount and type of precipitation is observed for the considered time period. Snowfall provided the majority of precipitation for 2009, whereas hoar frost and diamond dust were predominant in 2008 and 2010, respectively (Fig. 3).

The total annual accumulation derived from the precipitation measurements is 37-50% lower than the long-term average annual accumulation rate for Dome C (25 kg m$^{-2}$yr$^{-1}$) derived from firn core and stake measurements (Frezzotti et al., 2005, Urbini et al. 2008). Comparison between summer stake field measurements and cumulative accumulation from our samples also shows a low bias of 40% in our sampling. This is probably due to the loss of snow particles from our sampling plate, which may arise





from a lower cohesion forces between snow particles on the wooden plate compared to the forces within the snow, or to the higher wind speed at 1m height than directly at the surface, or to enhanced sublimation on the plate.

The ECMWF model displays a correct timing for large snowfall events and yields higher accumulation than the measurements (14.8, 17.9 and 12.4 mm w.e. for 2008, 2009 and 2010 respectively), but less than inferred from the stake array. Sublimation

is simulated by the model. It represents a minor fraction of snowfall (<10%), with net annual values of -1.4, -0.16 and -1.4 mm w.e. for 2008, 2009 and 2010, respectively, related to sublimation events of ~ -3 mm w.e during summer, partially countered by deposition events during winter.

The LMDZ model cumulative snowfall is fairly similar to the ECMWF, but this model produces more snowfall than ECMWF in 2008 and less than observed in 2009 and 2010. The ECMWF model seems to correctly reproduce the observed seasonal

accumulation variability. On the other hand, LMDZ seems to be able to reproduce only larger monthly accumulation peaks with greater discrepancies during late 2009 - early 2010 (not shown).

### 3.3 Observed $\delta^{18}O$ -temperature relationship

Least-square regression analysis of the relationship between $\delta^{18}O$ and temperature from daily or monthly observations (using

2 m and inversion temperature) and simulations (LMDZ) are reported in Table 2. Note that for better comparison with the data from other stations (Table 1) we did not remove seasonality from our data.

We first focus on the daily data. The $\delta^{18}O/T_{2m}$ slope is 0.49±0.02‰/°C ($R^2$=0.63, n=500), lower than the spatial slope of 0.8‰/°C obtained by Masson-Delmotte et al. (2008) for their database of Antarctic surface snow isotopic composition, which is itself close to the value expected from a Rayleigh distillation. Our results are intermediate between those obtained at other

East Antarctic plateau sites (Table 1). Based on one-year precipitation sampling in other inland Antarctic sites (Table 1), contrasted results were obtained at Vostok, where the associated slope (2.2‰/°C for $\delta D$, Ekaykin et al., 2004) was 3 times lower than the spatial one, while at Dome F, the $\delta^{18}O$ slope was close to the spatial relationship in daily precipitation samples collected in 2003 (0.78‰/°C, Fujita and Abe, 2006). However, Motoyama et al. (2005) reported a $\delta^{18}O$ slope of 0.57‰/°C based on monthly fresh snow samples collected in 1997 at Dome F. Investigations performed at coastal sites using either

shallow ice core data or precipitation data have systematically reported temporal slopes significantly lower than the average spatial relationship: a seasonal slope estimated at 0.44‰/°C for $\delta^{18}O$ from Law Dome ice cores (van Ommen and Morgan, 1997), inter-annual slopes of 0.24-0.34‰/°C for $\delta^{18}O$ from coastal Dronning Maud Land ice cores (Fernandoy et al., 2010) and a value of 0.57‰/°C for fresh snow samples collected at Neumayer Station during 1981-2006 (Schlosser et al., 2004).

Different results emerge from the $\delta^{18}O/T_{inv}$ relationship (Table 2), which leads to weaker correlation coefficients ($R^2$=0.44,

n=440) than for $T_{2m}$. The slope of the linear regression between $\delta^{18}O$ and $T_{inv}$ data is 0.84±0.05‰/°C, almost twice steeper than the one obtained with $T_{2m}$.



If we consider summer (defined as Nov-Feb) and winter (Mar-Oct) separately (Table 1), the relationship of $\delta^{18}$O and $T_{2m}$ is clearly different in winter and summer; we observe a slope of 0.59±0.03‰/°C ($R^2$=0.44, n=398) for $\delta^{18}$O/$T_{2m}$ in winter compared to 0.37±0.07‰/°C in summer ($R^2$=0.25, n=102). By contrast, summer and winter slopes are fairly similar for $\delta^{18}$O/$T_{inv}$, 0.59±0.06‰/°C for winter and 0.57±0.11‰/°C for summer, although with a lower regression coefficient ($R^2$=0.22 and 0.23, n=345 and 95, respectively).

When considering now monthly mean values (Table 2), correlation coefficients significantly increase, without any significant change in the regression slopes with respect to the results obtained from the daily data when considering the relationship with $T_{2m}$, but with a further increase of the slope when considering $T_{inv}$.

The obtained results are summarized in Figure 4 showing the $\delta^{18}$O /$T_{2m}$ relationship for both daily and monthly values as well as the linear regressions considering summer or winter precipitation. This figure clearly suggests that the precipitation occurring at temperatures above -40°C are characterized by a higher scatter and by a slightly lower δ/T slope.

We have also mimicked the record expected to be archived in ice cores, by considering annual precipitation weighted $T_{2m}$ and $\delta^{18}$O values (Table 3). Most of the precipitation-weighted annual mean temperature and $\delta^{18}$O (column c and f) are higher than the un-weighted annual averages (column b and e), because precipitation events are generally associated with warmer than average conditions. The difference between weighted and un-weighted isotopic and temperature values is greater during 2010 which is also the coldest year in the study period. When considering the linear regression between mean monthly weighted $T_{2m}$ and $\delta^{18}$O data (not shown), we obtain similar correlation coefficients and slopes. Again, the uncertainty associated with the sampling of individual precipitation events and therefore their weight does not allow to investigate further.

When considering inter-annual variations, we obtain a high $\delta^{18}$O/$T_{2m}$ slope (1.4±1.1‰/°C) for values calculated from the whole data set, as well as negative slopes for precipitation weighted values. However, their significance is very low due to the calculation limited to a three-year period. A further calculation that can be made in order to get rid of the mean seasonal cycle, which is dominating the statistical analysis, is to consider the mean monthly anomalies of temperature and $\delta^{18}$O for the study period. This leads to a higher $\delta^{18}$O/$T_{2m}$ slope of 0.96‰/°C ($R^2$=0.68) when compared to the one obtained from the mean monthly values (0.42‰/°C). However, also in this case its significance can be questioned due to the short period considered here.

The classification of each sample into snowfall, diamond dust, and hoar frost allowed us to investigate the relationship between the $\delta^{18}$O and temperature for the different snow types (Table 4 and 5). This investigation does not allow to identify any significant difference in the slopes between $\delta^{18}$O and $T_{2m}$ or $T_{inv}$ for the different categories. However, while diamond dust and snowfall show fairly similar isotopic compositions (Table 5), our data depict a specific isotopic fingerprint of hoar frost through more depleted $\delta^{18}$O and $\delta$D, and higher deuterium excess than for the other two types of precipitation.

Finally, we investigate the linear correlation between $\delta^{18}$O and $T_{2m}$ in the LMDZiso model, which demonstrates to capture quite well most of the precipitation events and their isotopic signature. The LMDZiso model underestimates the strength of



the linear correlation between $\delta^{18}O$ and $T_{2m}$, but with a slope close to that observed (slope of 0.46‰/°C, $R^2$=0.25 for daily data).

## 4 Discussion and conclusions

We provide the longest record of daily precipitation amounts and isotopic composition inferred from direct precipitation sampling from the central East Antarctic plateau, together with an analysis of precipitation types (diamond dust, snowfall, hoar frost) based on crystal typologies. This, for the first time, enabled us to study the relationship between local meteorological data and precipitation isotopic composition at intra-annual to inter-annual scale for the different types of precipitation. We observe a strong relationship between $\delta^{18}O$ ($\delta D$, not shown) and $T_{2m}$ or $T_{inv}$ at intra-seasonal to inter-annual scales (Table 2).

No noticeable differences are found in the slopes when considering daily samples or monthly values. Slopes of the $\delta^{18}O$ –T relationship are systematically twice higher for $T_{inv}$ than for $T_{2m}$, albeit with reduced correlation coefficients for inversion temperature.

Our results show an anti-phase between deuterium excess and $\delta^{18}O$, confirming earlier results from surface snow data (Masson-Delmotte et al., 2008) and the 1-yr-Dome Fuji precipitation data set (Fujita and Abe, 2006). Touzeau et al. (2016) reinforce

the view, already suggested by Uemura et al. (2012), that the main control on the precipitation deuterium excess in the coldest parts of inland Antarctica, at the very end of the distillation pathways, is indeed the site temperature. Deviations from such an anti-phase between deuterium excess and $\delta^{18}O$, as observed in Antarctic coastal areas, are expected to reflect changes in moisture source characteristics. We note that the LMDZiso model does capture this anti-phase at Concordia, but underestimates its strength, possibly due to the model warm bias.

Different precipitation types are expected to reflect different final condensation processes. Hoar frost represents inverse sublimation (deposition) of water vapour close to the snow surface, whereas snowfall and diamond dust form at height in the boundary layer. We identify an isotopic fingerprint of hoar frost through depleted $\delta^{18}O$ and $\delta D$, and higher deuterium excess. However, we cannot rule out that part of this depletion could be related to the fact that hoar frost occurs mainly during the coldest months. Based on the available data alone, it cannot be assessed whether this depletion arises from lower condensation

temperatures during hoar frost events or from condensation of an initially more depleted vapour. Both the combined monitoring of vapour and precipitation isotopic composition at Concordia and an exact determination of the moisture sources for diamond dust is needed to understand the processes at play.

This new Concordia data set can now be used to evaluate isotopic-enhanced atmospheric general circulation models. The comparison of our precipitation amount data with the ECMWF ERA Interim re-analysis shows good skills of the model.

Moreover, the capacity of the LMDZiso model to capture some of the observed $\delta^{18}O$ variance despite a warm bias and low resolution is encouraging. Our new dataset can be used for the evaluation of the model water cycle, for instance by investigating the sensitivity of model performance to spatial resolution or to different parameterizations.



The three-year monitoring of water stable isotopes of daily precipitation at Concordia station allowed to derive a $\delta^{18}O/T_{2m}$ slope of 0.49‰/°C, which is lower than the spatial slope of 0.8‰/°C obtained by the surface snow isotopic composition in Antarctica. This lower slope is also displayed in the LMDZiso simulation, suggesting that this model may capture correctly the distillation processes, despite its systematic biases. The weaker sensitivity of $\delta^{18}O$ to temperature presented here suggests

that the use of the spatial slope for temperature reconstructions in ice cores may lead to an underestimation of actual past temperature changes.

Our approach should be further expanded using observational atmospheric data, including vertical profiles of humidity and air temperature as well as vertical distribution of ice particles, in order to better define the temporal variability of the height of the condensation level. Furthermore, long-term (multi-decadal) monitoring is required to investigate the relationship between

large-scale patterns of the atmospheric circulation with the corresponding changes in moisture sources and transport paths and precipitation isotopic composition. Such monitoring will also provide a key reference for investigating post-depositional processes, which can alter the isotopic composition of surface snow. Recent studies performed in summer in Greenland and Antarctica have highlighted parallel shifts in surface vapour and surface snow isotopic composition in between snowfall events, possibly due to the uptake of surface vapour signals during snow metamorphism (Steen-Larsen et al., 2014; Ritter et al., 2016).

Moreover, Touzeau et al. (2016) analysed the $^{17}O$-excess of a subset of the 2010 precipitation samples from Concordia as well as surface snow samples and snow pits at Dome C and Vostok. They suggested that the isotope-temperature slope was even lower in surface snow samples than in precipitation samples. Again, this points to significant interactions of atmospheric water vapour and surface snow leading to post-depositional effects, potentially important in such arid regions of inland Antarctica. The continuous monitoring of the isotopic composition of precipitation, water vapour as well as surface snow will be

instrumental to investigate potential distortions in subsequent firn isotopic profiles.

**Author contribution**

BS, GD and MB are responsible for the precipitation measurements and stable isotope analysis, MV and AC for the crystal

analysis, CS, VC and PG for the meteorological and ECMW model data, CR for the LMDZ model data, DK collected the samples in the field. BS, CS, VC, VMD and ES prepared the manuscript. All the authors contributed to the data interpretation.

**Acknowledgements**

The precipitation measurements at Dome C as well as the isotopic analysis have been conducted in the framework of the

Concordia station glaciology and ESF PolarCLIMATE HOLOCLIP projects funded in Italy by PNRA-MIUR. This is a HOLOCIP publication number xx. Calculations, model comparison and analysis have been conducted in the framework of the MALOX and PRE-REC projects, funded by PNRA-MIUR. We appreciate the support of the University of Wisconsin-Madison Automatic Weather Station Program with the Dome C II data set. (NSF grant numbers ANT-0944018 and ANT-12456663) and the support of the IPEV/PNRA Project "Routine Meteorological Observation at Station Concordia with the radio sounding





data set. We thank all people who were involved in the precipitation sampling in the field, Daniele Frosini and Laura Genoni for the 2008 and 2009 sampling collection, respectively.

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





**Table 1.** Comparison of stable isotope – air temperature slopes for different sites in Antarctica.

| Site name | Lat. (°) | Long. (°) | Elevation (m) | δ¹⁸O/T slopes (‰/°C) | Statistics & samples | Time period | Type of data for isotopes analysis | Type of data for temperature analysis | Notes | References |
|---|---|---|---|---|---|---|---|---|---|---|
| Dome C | -75.12 | 123.31 | 3233 | 0.42-0.49 | | 2008-2010 (Daily and monthly values) | Precipitation samples | AWS | | this study |
| Dome F | -77.32 | 39.7 | 3810 | 0.78 | R=0.887 N=330 | 2003 (Daily values) | Precipitation samples | AWS | | Fujita and Abe (2006) |
| | | | | 0.52 | N=11 | 1997 (Monthly values) | Freshly fallen snow, drifting snow and surface snow samples | AWS | | Motoyama et al. (2005) |
| Vostok | -78.28 | 104.80 | 3488 | δD: 2.12 | | 2000 (monthly values) | Snow precipitation | AWS | | Ekaykin et al., 2004 |
| Law Dome | -66.77 | 112.98 | 1370 | 0.44 | R²=0.99 N=12 | Mean monthly values (Temperature 1986-1991, isotope 1304-1988) | Ice core | AWS | | van Ommen and Morgan (1997) |
| Neumayer Station | -70.65 | -8.25 | 42 | 0.57 | R=0.69 N=248 | 1981-2000 (Monthly values) | Fresh-snow samples | AWS | | Schlosser et al. (2004) |
| O'Higgins (Ant. Pen.) | -63.32 | -57.90 | 13 | 0.41 | R=0.79 N=180 | 2008-2009 (Monthly values) | Firn-cores | AWS | | Fernandoy et al. (2012) |
| Faraday St. | -65.25 | -64.27 | 0 | 0.46 | R=0.61 N=96 | 1964-1971 (Monthly values) | Ice-cores | 10 m depth temperature | | Peel et al. (1988) |





| | | | | | | | | | |
|---|---|---|---|---|---|---|---|---|---|
| Antarctic Peninsula | | | | 0.5-0.6 | R=0.3-0.5 N=21 | 1960-1980 (Annual values) | Ice-cores | 10 m depth temperature | Isotope at Gomez and Dolleman Temperature at Faraday and Halley Stations | |
| DML coastal area | | | | 0.24-0.34 | R= 0.54-0.71 N=300 | 1981-2006 | Firn-cores | AWS | Temperature at Neumayer Station | Fernandoy et al. (2010) |
| SW Ant. Pen | -73.59 | -70.36 | 1400 | 0.5 | n.a. | 1980-2005 | Ice core | 2m ECMWF temperature | Gomez ice core | Thomas et al. (2009) |
| Antarctica | | | | 0.34 (Continent) | $R^2$=0.25 | 2020-2100 | Model simulations | 2m HadAM3 temperature | HadAM3 simulations with an increased $CO_2$ 2100 climate projection scenario | Sime et al. (2008) |
| | | | | 0.47 (East Antarctica) | $R^2$=0.45 | | | | | |
| | | | | 0.5 (Peninsula) | $R^2$=0.56 | | | | | |




**Table 2.** $\delta^{18}$O-T (T$_{2m}$ and T$_{inv}$), d-$\delta^{18}$O, $\delta^{18}$O-$\delta^{18}$O$_{LMDZ}$, T$_{2m}$-T$_{2mLMDZ}$, d$_{LMDZ}$-$\delta^{18}$O$_{LMDZ}$ and $\delta^{18}$O$_{LMDZ}$-T$_{2mLMDZ}$ regressions for daily and monthly precipitation at Concordia, over the period 2008-2010.

| Variables (Y/X) | Slope | Intercept | $R^2$ | N sample | Sampling (Period) |
|---|---|---|---|---|---|
| $\delta^{18}$O/T$_{2m}$ | 0.49±0.02 | -30.79±0.97 | 0.63 | 500 | Daily (all sample) |
| $\delta^{18}$O/T$_{2m}$ | 0.37±0.07 | -35.33±2.16 | 0.25 | 102 | Daily (NDJF) |
| $\delta^{18}$O/T$_{2m}$ | 0.59±0.03 | -24.97±2.05 | 0.44 | 398 | Daily (MAMJJASO) |
| $\delta^{18}$O/T$_{2m}$ | 0.42±0.04 | -33.94±1.93 | 0.82 | 33 | Monthly (all sample) |
| $\delta^{18}$O/T$_{INV}$ | 0.84±0.05 | -25.74±1.76 | 0.44 | 440 | Daily (all sample) |
| $\delta^{18}$O/T$_{INV}$ | 0.57±0.11 | -30.50±3.34 | 0.23 | 95 | Daily (NDJF) |
| $\delta^{18}$O/T$_{INV}$ | 0.59±0.06 | -37.113±2.41 | 0.22 | 345 | Daily (MAMJJASO) |
| $\delta^{18}$O/T$_{INV}$ | 1.16±0.10 | -13.20±3.80 | 0.81 | 33 | Monthly (all sample) |
| d/$\delta^{18}$O | -1.49±0.04 | -68.80±2.57 | 0.70 | 499 | Daily (all sample) |
| d/$\delta^{18}$O | -1.36±0.17 | -61.23±9.41 | 0.68 | 33 | Monthly (all sample) |
| $\delta^{18}$O/ $\delta^{18}$O$_{LMDZ}$ | 0.59±0.03 | -31.18±1.44 | 0.50 | 366 | Daily (all sample) |
| $\delta^{18}$O/ $\delta^{18}$O$_{LMDZ}$ | 0.84±0.09 | -18.60±4.00 | 0.75 | 33 | Monthly (all sample) |
| d/d $_{LMDZ}$ | 0.77±0.08 | 20.80±0.74 | 0.20 | 365 | Daily (all sample) |
| d/d $_{LMDZ}$ | -1.09±0.21 | -33.50±9.55 | 0.46 | 33 | Monthly (all sample) |
| T$_{2m}$/T$_{2mLMDZ}$ | 1.20±0.02 | 4.56±0.97 | 0.76 | 1110 | Daily (all sample) |
| T$_{2m}$/T$_{2mLMDZ}$ | 1.27±0.073 | 7.76±3.50 | 0.90 | 36 | Monthly (all sample) |
| d $_{LMDZ}$ /$\delta^{18}$O $_{LMDZ}$ | -0.52±0.02 | -27.60±1.03 | 0.43 | 726 | Daily (all sample) |
| d $_{LMDZ}$ /$\delta^{18}$O $_{LMDZ}$ | -0.38±0.07 | -21.19±3.22 | 0.45 | 36 | Monthly |
| $\delta^{18}$O$_{LMDZ}$/ T$_{2mLMDZ}$ | 0.46±0.03 | -23.11±1.48 | 0.25 | 726 | Daily (all sample) |





| | | | | | |
|---|---|---|---|---|---|
| $\delta^{18}O_{LMDZ}/$ $T_{2mLMDZ}$ | 0.53±0.06 | -19.34±2.91 | 0.69 | 36 | Monthly |
| $\delta D_{LMDZ}/\delta^{18}O_{LMDZ}$ | 7.48±0.02 | -27.60±1.03 | 0.99 | 726 | Daily (all sample) |

**Table 3.** Mean annual values for 2008, 2009 and 2010 of 2m-temperature and $\delta^{18}O$, considering the whole dataset (a and d column, respectively), a subset of data considering days when precipitation was quantified (b and e, respectively) and the whole dataset weighed by the precipitation amount (c and f, respectively).

| Year | a | b | c | d | e | f |
|---|---|---|---|---|---|---|
| | $T_{2m}$ simple annual average (°C) | $T_{2m}$ annual average based on precipitated days (°C) | $T_{2m}$ annual average weighed by precipitation (°C) | $\delta^{18}O$ simple annual average (‰) | $\delta^{18}O$ annual average based on precipitated days (‰) | $\delta^{18}O$ annual average weighed by precipitation (‰) |
| **2008** | -51.69 | -56.81 | -53.54 | -55.26 | -55.52 | -52.70 |
| **2009** | -49.88 | -55.63 | -54.03 | -55.47 | -55.86 | -53.20 |
| **2010** | -52.88 | -53.29 | -47.78 | -60.16 | -57.78 | -53.73 |



**Table 4.** $\delta^{18}O$ - T ($T_{2m}$ and $T_{inv}$) and d-$\delta^{18}O$ regressions for daily and monthly precipitation at Concordia, over the period 2008-2010 considering different precipitation types (snowfall, diamond dust and hoar frost).

| Variables (Y/X) | | Slope | Intercept | $R^2$ | N sample | Sampling (Period) |
|---|---|---|---|---|---|---|
| $\delta^{18}O/T_{2m}$ | Snowfall | 0.45±0.04 | -31.21±2.11 | 0.60 | 98 | Daily (all sample) |
| | Hoar frost | 0.48±0.04 | -29.84±2.69 | 0.44 | 159 | |
| | Diamond dust | 0.47±0.03 | -31.76±1.41 | 0.71 | 105 | |
| $\delta^{18}O/T_{INV}$ | Snowfall | 0.91±0.08 | -21.89±3.00 | 0.60 | 88 | Daily (all sample) |
| | Hoar frost | 0.49±0.08 | -40.26±3.32 | 0.21 | 133 | |
| | Diamond dust | 0.83±0.08 | -23.35±2.74 | 0.55 | 99 | |
| d/$\delta^{18}O$ | Snowfall | -1.20±0.10 | -52.91±5.56 | 0.60 | 98 | Daily (all sample) |
| | Hoar frost | -1.32±0.08 | -57.63±4.67 | 0.65 | 158 | |
| | Diamond dust | -1.66±0.13 | -76.85±7.04 | 0.61 | 105 | |
| $\delta^{18}O/T_{2m}$ | Snowfall | 0.41±0.03 | -34.58±1.69 | 0.86 | 29 | Monthly (all sample) |
| | Hoar frost | 0.51±0.06 | -28.68±3.38 | 0.77 | 25 | |
| | Diamond dust | 0.41±0.04 | -34.67±1.84 | 0.84 | 28 | |
| $\delta^{18}O/T_{INV}$ | Snowfall | 1.10±0.10 | -15.13±3.46 | 0.84 | 29 | Monthly (all sample) |
| | Hoar frost | 1.34±0.15 | -6.83±5.66 | 0.78 | 25 | |
| | Diamond dust | 1.10±0.11 | -15.14±3.96 | 0.80 | 28 | |
| d/$\delta^{18}O$ | Snowfall | -1.30±0.208 | -57.91±10.842 | 0.63 | 29 | Monthly (all sample) |
| | Hoar frost | -0.92±0.16 | -37.19±9.74 | 0.60 | 25 | |
| | Diamond dust | -1.38±0.20 | -62.57±11.08 | 0.65 | 28 | |





**Table 5.** Mean, minimum and maximum isotopic values ($\delta$D, $\delta^{18}$O, deuterium excess) for the different precipitation types over the considered three-year period.

| Isotopic values | | Snowfall | Diamond Dust | Hoar Frost |
|---|---|---|---|---|
| **$\delta$D** | *mean* | -431.9 | -411.5 | -453.5 |
| | *min* | -530.2 | -542.9 | -574.1 |
| | *max* | -284.0 | -297.2 | -328.7 |
| **$\delta^{18}$O** | *mean* | -55.70 | -52.79 | -59.26 |
| | *min* | -70.69 | -73.80 | -79.11 |
| | *max* | -37.32 | -36.27 | -41.04 |
| **deuterium excess** | *mean* | 13.7 | 10.8 | 20.6 |
| | *min* | -16.5 | -49.6 | -7.5 |
| | *max* | 37.3 | 47.7 | 58.8 |





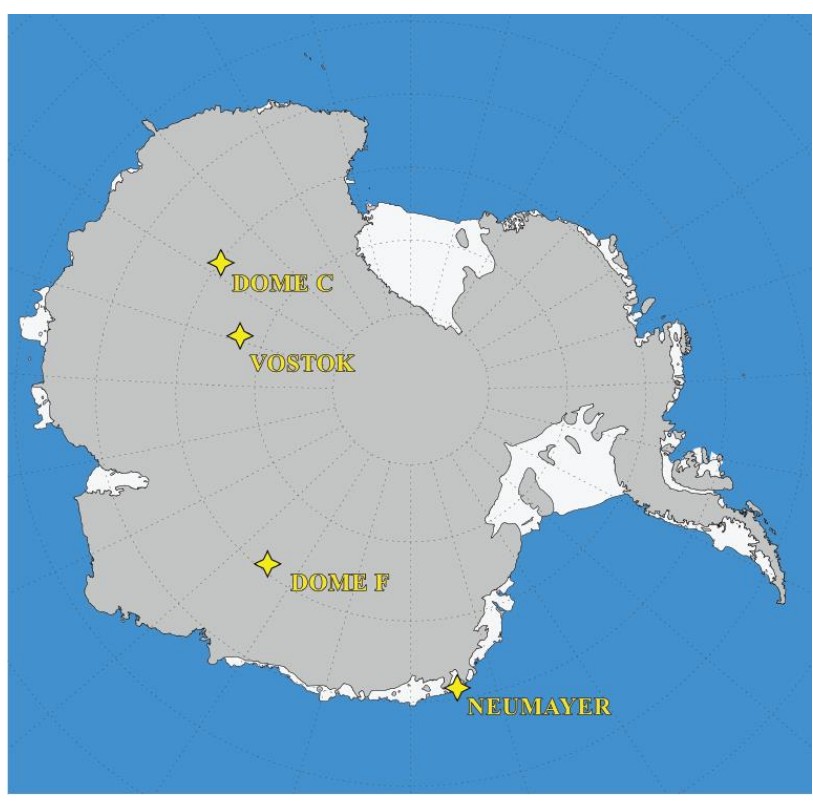

5 **Figure 1.** Map of Antarctica showing the location of the study site Concordia station at Dome C and other sites cited in the text.





**Figure 2.** Meteorological variables and stable isotope data of fresh snow samples from Concordia site for the 2008-2010 study period:





a) Daily mean 2 m air temperature from AWS ($T_{2m}$, blue line), temperature at the top of the inversion layer ($T_{inv}$ from radiosounding profiles, dark green line), and $\delta^{18}O$ of precipitation samples (filled orange circle). Monthly means of $T_{2m}$, $T_{inv}$ and $\delta^{18}O$ are given as horizontal double red, green and yellow lines, respectively.

b) Deuterium excess of the daily precipitation samples.

c) Cumulative measured precipitation amount (black line), measured cumulative precipitation for diamond dust (green line), snowfall (red line) and hoar frost (blue line), and modelled cumulative precipitation from ECMWF (snowfall, snowfall minus evaporation, orange and light blue, respectively) and LMDZ (snowfall, light grey line). Cumulative accumulation from stake array (dark grey vertical bars). Stake data are only available in the respective summer months.




**Figure 3**. Monthly totals of precipitation for 2008-2010, given for diamond dust, hoar frost, and snowfall (a) and relative amounts (%) of diamond dust, hoar frost, and snowfall for each year (b, c, d) and for the entire period 2008-2010 (e).







**Figure 4.** $\delta^{18}O$ values, measured on daily collected sample, with respect to daily average 2-meter air temperature (orange filled circles). Dark red filled square and error bars represent their monthly average and standard deviation respectively. Linear regression fits of daily ($Y=-30.79+0.49*X$, $R^2=0.63$, $N=500$) and monthly ($Y=-33.94+0.42*X$, $R^2=0.82$, $N=33$) data are shown in yellow and red thick dashed lines, respectively. Monthly linear regression using only November-February (NDJF, $Y=-45.33+0.072*X$, $R^2=0.05$, $N=9$) and March-October (MAMJJASO, $Y=-18.46+0.68*X$, $R^2=0.65$, $N=24$) are highlighted in green and light blue thick dashed lines, respectively. Black thin line represents the one to one line.