# Peer review of "Three-year monitoring of stable isotopes of precipitation at Concordia Station, East Antarctica"

_The Cryosphere, 2016_

## Referee Comment (RC1) · Anonymous Referee #1 · 2 Jul 2016

The manuscript summarizes data on isotopic composition of snow precipitation collected on daily basis in central Antarctica during 3 complete years. The collected data are unique, there are very few datasets like this for central Antarctica. The isotopic content data are interpreted in terms of air temperature using AWS temperature observations, as well as GCM modeling. The results rise many questions concerning the isotope-temperature relationships and, in general, the factors that form the isotopic composition of precipitation. I believe these question will be addressed in future papers. The results of this study, combined with the ongoing studies of isotopic composition of atmospheric water vapor and post-depositional processes in snow, will contribute a lot to understanding the deep ice core isotopic composition data.

Specific comments:

page 5, line 28 and page 6, line 1: please indicate, what was the sampling frequency before end of 2007.

page 6, line 11: or supersaturated air mass?

page 8, lines 13-14: should average slope (0.35) be between summer (0.51) and winter (0.39) slopes?

page 9, line 18: do these precipitation amounts account for the hoar frost?

page 9, line 24: I cannot see the number of snowfall events in Figure 3.

page 9, line 31: it's better to use the same units for precipitation and accumulation, for example, mm w.e.

page 10, lines 5-7: I did not understand well this part of text. Annual sums of sublimation are 1.4, 0.16 and 1.4 mm, right? Sublimation occurs in summer, and summer sublimation is 3 mm, right? Then the difference between summer sublimation (3 mm) and total annual sublimation (e.g., 1.4 mm) is due to partial re-deposition of moisture during the winter months?

The same place: when you write "sublimation = -3 mm" for me it means that the water vapor is re-sublimated on the snow surface, since negative sublimation = re-sublimation or deposition.

Concerning difference between measured precipitation amount and accumulation: can it be due to underestimation of the hoar frost deposition that is likely an important contributor to mass balance at Dome C?

page 11, lines 26-30: do you have the data on snow surface temperature to compare with the hoar isotopic composition? For the hoar, condensation temperature should be close to snow surface temperature (or air temperature just above the snow) that may be significantly lower than the T2m.

Table 1: at Vostok, temperature was obtained not at AWS, but from the "classical"

manned meteorological station by manual measurements.

Table 24: first 3 lines show isotope-T2m slopes, while lines 4-6 show slope of isotopes versus inversion temperature. Thus in lines 1-3 the slope values are smaller, because the amplitude of T2m variability is much larger than that of the inversion temperature. But why value in line 2 is almost the same as in line 5?

—————————————————————

---

## Referee Comment (RC2) · Anonymous Referee #2 · 23 Jul 2016

Evaluation of Manuscript General comments: This paper showed daily oxygen and hydrogen isotope compositions of snow over three years at EDC station, Antarctica. The data provide fundamentally important information to interpret the climate records from the Antarctic ice cores. The authors showed that temporal slope of d18O vs 2-m air temperature is smaller than the spatial one, which is generally consistent with previous a few observations in other sites. They also categorize the snow based on crystal morphologies. Overall, the manuscript is easy to understand, and provides very precious data set. I recommend accepting the paper after the authors address minor comments below.

Specific comments:

P6 L. 18-19: The precipitation. . .. were examined daily using a magnifying lens and a

high-resolution camera to determine these three board types of precipitation (snowfall, diamond dust, and hoar frost).

- Please show typical photographs showing the three types of snow crystals. Such information is necessarily to evaluate the validity of the categorization and to conduct a similar observation in aother Antarctic site.

P11 L. 11: characterized by a higher scatter and by slightly lower d/T slope

- I agree it seems to show a lower slope, but I cannot see a higher scatter. In addition, the temperature at which the slope changes appears to be around -50 degC.

Technical corrections:

P7 3.1. Observed and simulated variability of temperature...:

- Observed and simulated variabilities?

Figure 2

- Please add precise explanation for the each panels, axes, lines and symbols in figure caption. The legends in the figs are vague and difficult to read.

- Figure 2c would be separated from Fig. 2a and Fig. 2b. Fig. 2c is not directlly relevant to Figs. 2a and 2b.

---

## Author Response (AR1)

**Authors' response:**

We thank the reviewers for their comments. Below, the reviewers' comments are in black, our responses to the individual comments are displayed in red and suggested modifications in the manuscript in blue (changes in sentences are highlighted in bold).

**Anonymous Referee #1**

The manuscript summarizes data on isotopic composition of snow precipitation collected on daily basis in central Antarctica during 3 complete years. The collected data are unique, there are very few datasets like this for central Antarctica. The isotopic content data are interpreted in terms of air temperature using AWS temperature observations, as well as GCM modeling. The results rise many questions concerning the isotope-temperature relationships and, in general, the factors that form the isotopic composition of precipitation. I believe these question will be addressed in future papers. The results of this study, combined with the ongoing studies of isotopic composition of atmospheric water vapor and post-depositional processes in snow, will contribute a lot to understanding the deep ice core isotopic composition data.

Specific comments:

page 5, line 28 and page 6, line 1: please indicate, what was the sampling frequency before end of 2007.

Before the end of 2007 the samples were collected only when the snow accumulated over the wooden platform reached a defined threshold height (generally every four – five days), but that frequency was not satisfactory in order to decouple variability of different phenomena affecting snow accumulation and isotopes and so these data have not been included and used only for testing the observation system.

page 6, line 11: or supersaturated air mass?

In this sentence we used the expression "an almost saturated air mass" considering a wide category of atmospheric conditions leading to the formation of diamond dust. Anyway, following the suggestion, we changed the sentence as follow and we added a reference: " It consists of very fine ice crystals that form due to radiative cooling of an almost saturated or supersaturated air mass. **Also mixing of moist warmer air with colder air can lead to supersaturation of the cold air and thus formation of ice crystals (Walden et al., 2003).** "

Reference:

Walden, V.P., Warren, S.G., Tuttle, E.: Atmospheric Ice Crystals over the Antarctic Plateau in Winter, J. Appl. Meteor. 42 (10), 1391-1405, doi:10.1175/1520-0450(2003)042<1391:AICOTA>2.0.CO;2, 2003.

page 8, lines 13-14: should average slope (0.35) be between summer (0.51) and winter (0.39) slopes?

No, because it is not an average slope. We do not apply regression to winter and summer sub-dataset and then average the resulting slopes but the value 0.35 represents a different regression over the entire daily

dataset (1002 points). Moreover, summer and winter sub dataset are not composed by an identical number of points (345 and 657 samples, respectively).

page 9, line 18: do these precipitation amounts account for the hoar frost?

Yes, the precipitation amounts account for the hoar frost. We changed the text in order to improve readability as follows:

"The annual amounts of **the sample**d snow **(snowfall + diamond dust + hoar frost)** are ..."

page 9, line 24: I cannot see the number of snowfall events in Figure 3.

We added the numbers and the total absolute amounts of different precipitation events in the pie plots in figure 3, in order to improve readability as shown below.

[Figure]

Figure 5. Monthly totals of precipitation for 2008-2010, given for diamond dust, hoar frost, and snowfall (a); **relative (%) and absolute amounts (mm w.e.) and number of events** for diamond dust, hoar frost, and snowfall for each year (b, c, d) and for the entire period 2008-2010 (e).

page 9, line 31: it's better to use the same units for precipitation and accumulation, for example, mm w.e.

We followed the suggestion and we changed the text accordingly.

page 10, lines 5-7:  I did not understand well this part of text.  Annual sums of sublimation are 1.4, 0.16 and 1.4 mm, right?  Sublimation occurs in summer, and summer sublimation is 3 mm, right?  Then the difference between summer sublimation (3 mm) and total annual sublimation (e.g., 1.4 mm) is due to partial re-deposition of moisture during the winter months?

The same place: when you write "sublimation = -3 mm" for me it means that the water vapor is re-sublimated on the snow surface, since negative sublimation = re-sublimation or deposition.

The referee understands perfectly the meaning of the sentence. We obtained the model sublimation values from the ECMWF model evaporation field. It represents the turbulent moisture flux exchange between surface and atmosphere. It is defined positive downward (from atmosphere toward surface), when moisture condenses and reaches the ground. On the other hand, it is negative upward (from surface toward atmosphere) when solid snow sublimates into gaseous form moving toward first atmosphere layers. We used inappropriately the same name (sublimation) in order to describe both the model evaporation vector and the state passage. We removed the minus sign to all the sublimation values in order to improve readability.

Concerning difference between measured precipitation amount and accumulation: can it be due to underestimation of the hoar frost deposition that is likely an important contributor to mass balance at Dome C?

The hoar frost deposition is surely an important contributor to the surface mass balance at Dome C. Our datasets clearly indicate that on average it weighs more than 30% of the total accumulation over the wooden platform, with few variations during the three sampling years. Under the extreme conditions occurring in the East Antarctic plateau the very large accumulation spatial variability (Frezzotti et al 2007) is enhanced (Genthon et al 2015) and small effects like hoar frost deposition could be magnified or completely suppressed by very local scale conditions (e.g snow deposited over plate blown away by the wind).

page 11, lines 26-30: do you have the data on snow surface temperature to compare with the hoar isotopic composition? For the hoar, condensation temperature should be close to snow surface temperature (or air temperature just above the snow) that may be significantly lower than the T2m.

Unfortunately, we do not have measurements of the surface temperature. We could calculate a daily surface temperature time series starting from the longwave downward radiation data but those data are available only for 2010 and the calculation would be affected by large uncertainties related to snow conditions at the ground.

Table 1: at Vostok, temperature was obtained not at AWS, but from the "classical" manned meteorological station by manual measurements.

We changed the table following the suggestion.

Table 24: first 3 lines show isotope-T2m slopes, while lines 4-6 show slope of isotopes versus inversion temperature. Thus in lines 1-3 the slope values are smaller, because the amplitude of T2m variability is much larger than that of the inversion temperature. But why value in line 2 is almost the same as in line 5

This is probably related to the fact that the hoar events occur mainly during wintertime when the atmosphere is stable and stratified and the T2m variability is more limited and similar to the one at inversion height. On the other hand, snowfall and diamond dust occur over the whole year when the atmosphere is less stable and more mixed and the $T_{2m}$ variability is higher than at the inversion level.

We added a sentence at page 11, line 28 in order to explain this discrepancy raised by the referee. Now the paragraph is changed as below:

The classification of each sample into snowfall, diamond dust, and hoar frost allowed us to investigate the relationship between the $\delta^{18}O$ and temperature for the different snow types (Table 4 and 5). This investigation does not allow to identify any significant difference in the slopes between $\delta^{18}O$ and $T_{2m}$ or $\delta^{18}O$ and $T_{inv}$ within the different categories, **with the exception of hoar frost which shows similar slope values for both $T_{2m}$ and $T_{inv}$. This is probably related to the fact that the hoar events occur mainly during wintertime when the atmosphere is stably stratified and the $T_{2m}$ variability is more limited and similar to the one at inversion height. Moreover**, while diamond dust and snowfall show fairly similar isotopic compositions (Table 5), our data depict a specific isotopic fingerprint of hoar frost through more depleted $\delta^{18}O$ and $\delta D$, and higher deuterium excess than for the other two types of precipitation.

**Anonymous Referee #2**

Evaluation of Manuscript General comments: This paper showed daily oxygen and hydrogen isotope compositions of snow over three years at EDC station, Antarctica. The data provide fundamentally important information to interpret the climate records from the Antarctic ice cores. The authors showed that temporal slope of d18O vs 2-m air temperature is smaller than the spatial one, which is generally consistent with previous a few observations in other sites. They also categorize the snow based on crystal morphologies. Overall, the manuscript is easy to understand, and provides very precious data set. I recommend accepting the paper after the authors address minor comments below.

Specific comments:

P6 L. 18-19: The precipitation.... were examined daily using a magnifying lens and a high-resolution camera to determine these three board types of precipitation (snowfall, diamond dust, and hoar frost).

- Please show typical photographs showing the three types of snow crystals. Such information is necessarily to evaluate the validity of the categorization and to conduct a similar observation in aother Antarctic site.

We added a figure (see below) with examples of crystal types for each category (citation at page 6, line 19). We added a figure caption (as below) and one reference in the manuscript.

[Figure]

**Figure 2. Photographs of precipitation types as identified on the wooden platform according to the Magono and Lee (1966) classification. a) hoar frost; b) snowfall: combinations of bullets (C2a); c) mixture of minute hexagonal plate (diamond dust, G3) and thick plate of skeleton form (snowfall, C1h) crystals. The photographs have been performed by the winter over personnel at Concordia station**.

**Reference**

**Magono, C. and Lee, C. W.: Meteorological Classification of Natural Snow Crystals, Journal of the Faculty of Science, Hokkaido University, Series 7, Geophysics, 2(4), 321-335, 1966.**

P11 L. 11: characterized by a higher scatter and by slightly lower d/T slope

- I agree it seems to show a lower slope, but I cannot see a higher scatter. In addition, the temperature at which the slope changes appears to be around -50 degC.

We changed the text accordingly to the suggestion.

"This figure clearly suggests that the precipitation occurring at temperatures above -50°C are characterized by a slightly lower $\delta$/T slope."

Technical corrections:

P7 3.1. Observed and simulated variability of temperature...:

- Observed and simulated variabilities?

Done (also for paragraph 3.2)

Observed and simulated variabilities of temperature and stable isotopes.

Observed versus simulated variabilities of precipitation and accumulation

Figure 2

- Please add precise explanation for the each panels, axes, lines and symbols in figure caption. The legends in the figs are vague and difficult to read.

- Figure 2c would be separated from Fig. 2a and Fig. 2b. Fig. 2c is not directly relevant to Figs. 2a and 2b.

Following the suggestions of reviewer, we split the figure in two parts and we tried to improve the figure captions, legends and symbols.

[revised manuscript text omitted]